# Outcomes and Patient Selection in Laparoscopic vs. Open Liver Resection for HCC and Colorectal Cancer Liver Metastasis

**DOI:** 10.3390/cancers15041179

**Published:** 2023-02-12

**Authors:** Jurgis Alvikas, Winifred Lo, Samer Tohme, David A. Geller

**Affiliations:** 1Department of Surgery, University of Pittsburgh Medical Center, Pittsburgh, PA 15213, USA; 2Division of Hepatobiliary and Pancreatic Surgery, University of Pittsburgh Medical Center, 3459 Fifth Avenue, MUH 7S, Pittsburgh, PA 15213, USA

**Keywords:** laparoscopic liver resection, laparoscopic hepatectomy, open liver resection, hepatocellular carcinoma, HCC, colorectal cancer liver metastasis, CRLM

## Abstract

**Simple Summary:**

Surgically removing part of the liver is an essential method of treating cancers in the liver. Two most commonly removed cancers are called hepatocellular carcinoma (HCC) and colorectal liver metastasis (CRLM). Over the last two decades, a minimally invasive technique, called laparoscopic liver resection (LLR), have been developed to make recovery from these operations easier compared to existing method called open liver resection (OLR). In this article, we review the studies that compared LLR and OLR and describe their findings. Patients undergoing LLR have fewer complications, reduced blood loss during the operation and shorter hospital length of stay with similar long-term survival as compared to OLR. We also describe an approach to selecting patients best suited for LLR and review literature behind a new emerging robotic-assisted liver resection technique.

**Abstract:**

Hepatocellular carcinoma (HCC) and colorectal liver metastasis (CRLM) are the two most common malignant tumors that require liver resection. While liver transplantation is the best treatment for HCC, organ shortages and high costs limit the availability of this option for many patients and make resection the mainstay of treatment. For patients with CRLM, surgical resection with negative margins is the only potentially curative option. Over the last two decades, laparoscopic liver resection (LLR) has been increasingly adopted for the resection of a variety of tumors and was found to have similar long-term outcomes compared to open liver resection (OLR) while offering the benefits of improved short-term outcomes. In this review, we discuss the current literature on the outcomes of LLR vs. OLR for patients with HCC and CRLM. Although the use of LLR for HCC and CRLM is increasing, it is not appropriate for all patients. We describe an approach to selecting patients best-suited for LLR. The four common difficulty-scoring systems for LLR are summarized. Additionally, we review the current evidence behind the emerging robotically assisted liver resection technology.

## 1. Introduction

Hepatocellular carcinoma (HCC) and colorectal cancer liver metastasis (CRLM) are the two most common malignant tumors requiring a liver resection [1]. Primary liver cancer is the third leading cause of cancer-related death in the world and comprises HCC (approximately 85%) and intrahepatic cholangiocarcinoma (approximately 15%) [2]. It typically occurs in patients with underlying liver cirrhosis, and surgical resection offers long-term survival benefits. However, due to an increased risk of HCC formation in the cirrhotic liver, about half of these patients will develop another HCC after resection [3]. While liver transplantation is the best treatment for HCC, it is limited by organ availability, high cost, and strict transplantation criteria. In addition, in 2015, UNOS mandated that all patients diagnosed with HCC wait 6 months before accruing the HCC MELD exception points [4]. As a result, the dropout rate for patients with HCC awaiting a liver transplant is currently as high as 29.0% based on national data [5]. Liver resection, therefore, is a critical tool for HCC management for many patients.

Colorectal cancer is the third most common cancer worldwide [2]. In the last several decades, significant advances in imaging, surgical techniques, and chemotherapeutic regimens have improved survival. For patients with CRLM, complete surgical resection is currently the only potentially curative option. Despite advances in chemotherapy and multi-modality approaches to colorectal cancer, the 5-year survival of patients with metastatic colorectal cancer is less than 20% [6], and liver metastasis resection is the best option to achieve long-term survival and/or cure the disease. 

Laparoscopic liver resection (LLR) has been shown to have many short-term advantages and patient benefits [1,7] compared to open liver resection (OLR) because of its better post-operative recovery while leading to similar long-term oncologic outcomes. Over the last two decades, LLR was shown to be successful for a variety of tumor types, for both minor and major liver resections, and in patient populations with significant comorbidities, such as advanced liver disease. The data on LLR have continued to evolve and multiple retrospective studies as well as several prospective randomized, controlled trials have corroborated these findings. LLR is now considered to number among the standard-of-care regimens for many patients requiring a liver resection [8,9]. 

In this review, we summarize the current literature on the outcomes of LLR vs. OLR with respect to HCC and CRLM. We include retrospective and prospective studies, highlight the landmark randomized controlled trials, and describe the emerging data on the robotic approach to liver resection. Although LLR is a broadly accepted technique, it is not appropriate in every case, and surgeons need to be within their comfort zone when selecting cases for LLR (which vary in terms of the degree of difficulty) based on their experience and training. We outline our approach to selecting the patients appropriate for LLR vs. OLR.

## 2. Methods

We performed a literature review of studies published in PubMed database from years 2001–2022 and used the following search terms: “laparoscopic liver resection and meta-analysis”. We excluded studies on benign disease, as well as studies that focused on malignancies other than HCC and CRLM. Among the studies identified, we performed a literature review focusing on the outcomes of laparoscopic liver resections regarding CRLM and HCC. For Table 1 and Table 2, we excluded studies that did not include 5-year overall survival (OS). 

## 3. Results

The majority of the studies indicate that LLR had better short-term outcomes than OLR and equivalent long-term oncologic outcomes. We included 17 studies on HCC (Table 1) and 16 studies on CRLM (Table 2). We summarize four commonly used difficulty-scoring systems (Table 3). Our search of meta-analyses on LLR identified 224 search results, of which we include 95 meta-analyses that have focused on the outcomes of LLR and compared it to OLR or RLR in different clinical settings (Table 4).

### 3.1. Hepatocellular Carcinoma (HCC)

In a large propensity score-matched analysis, Takahara et al. compared the short- and long-term outcomes of 387 patients with HCC after LLR and after OLR [13]. The groups had similar demographic characteristics, an average tumor size of 2.8 cm, and approximately 10% of the patients in each group had undergone a major hepatectomy. The authors found that the LLR group had significantly less intraoperative blood loss (158 g vs. 400 g, *p* < 0.001), shorter lengths of stay (13 vs. 16 days, *p* < 0.001), and fewer complications (6.7% vs. 13.0%, *p* = 0.003) but significantly longer operation times (294 vs. 271 min, *p* = 0.025). Importantly, there were no significant differences between the LLR vs. OLR groups in terms of 5-year Overall Survival (OS) (76.8% vs. 70.9%) or Disease-Free Survival (DFS) (40.7% vs. 39.3%).

A randomized controlled trial conducted by El-Gendi and colleagues aimed to compare the surgical and oncologic outcomes of LLR vs. OLR among patients with small (<5.0 cm) HCC and Child’s A cirrhosis [10]. The primary endpoint was postoperative length of stay (LOS), and 25 patients were assigned to each group. All resections were non-anatomical. The authors reported a significantly shorter LOS for the LLR group compared to OLR (2.4 vs. 4.3 days, *p* < 0.001). They also noted that the LLR group had shorter operation times (120 vs. 147 min, *p* < 0.001), earlier resumption of regular diet (1.1 vs. 2.8 days, *p* = 0.001), and shorter time requiring intravenous narcotic use (1.0 vs. 2.8 days, *p* < 0.001). The 3-year disease-free survival was 58.7% and 54.0% for the LLR and OLR groups, respectively.

A 2017 meta-analysis pooled data from 44 studies that investigated the outcomes of LLR vs. OLR in patients with HCC [43]. A total of 5203 patients were included in this analysis. The meta-analysis showed that LLR leads to less blood loss, lower postoperative pain levels, shorter hospital stay, better R0 resection rates, and better resection margins, although the last two findings were likely obtained due to the fact that the mean tumor size was 0.6 cm smaller in the LLR group. Similar to the studies noted above, the long-term oncological results (both OS and DFS) of LLR and OLR were found to be similar.

Another recent meta-analysis analyzed the LLR outcomes among patients with HCC and cirrhosis [44]. It sought to account for selection bias by including only randomized control trials and propensity-score-matched studies. To exclude the potential effects of the steep initial learning curve regarding LLR, only studies from 2015 to 2020 were included. The 5-year OS was 68.8% and 64.2% for the LLR and OLR groups, respectively. R0 resection rates were not different. The authors concluded that LLR leads to a survival advantage that persists over time. The LLR group also had fewer postoperative complications, shorter lengths of stay, and less intraoperative blood loss.

The majority of the reports of LLR for patients with HCC IIed patients with Child’s A cirrhosis. However, several recent studies have expanded LLR to patients with Child’s B cirrhosis. In a single-center study, Beard et al. examined whether LLR could be safely utilized among patients with Child’s B and Child’s C cirrhosis [45]. Eighty patients with HCC and Child’s A cirrhosis were compared to twenty-six patients with advanced cirrhosis. The primary outcomes of 30- and 90-day mortality as well as postoperative complications, R0 resection rate, and length of stay were not different between the two groups. The postoperative bilirubin increase was higher in patients with advanced cirrhosis. This study suggested that there may be patients with HCC and advanced cirrhosis who are candidates for LLR. A recent multi-institutional, propensity-score-matched analysis also found that LLR among patients with HCC and Child’s B cirrhosis has acceptable oncological outcomes and may confer benefits with less intraoperative blood loss, fewer complications, and shorter hospital stay [46]. 

While some pioneering experts in the field have reported that LLR is possible for tumors in all segments, certain tumor locations remain challenging. A study describing a technique for laparoscopic segmentectomies found that tumors in segments 1, 4a, 7, and 8 are particularly technically complex, result in longer operation times and increased blood loss [47], and should be performed by surgeons with advanced experience in both OLR and LLR. Additionally, laparoscopic posterosuperior segmentectomy and right posterior sectionectomy are recognized to have increased morbidity and higher complication rates [48]. A recent study identified parenchyma-sparing resections in segment 7 and the segmentectomy of segment 8 to be particularly technically challenging. However, these difficulties can be managed with lateral positioning or additional trocars, and patients undergoing LLR of lesions in segments 7 and 8 have less blood loss and shorter length of stay compared to OLR [49]. 

Lastly, while evidence regarding LLR among patients with HCC and cirrhosis has been growing, data on repeat LLR in this setting is limited. Morise and colleagues reported the outcomes of repeat LLR among patients with HCC [50]. They found that repeat LLR led to less intraoperative blood loss compared to OLR but longer operation times. Postoperative morbidity and OS were not different between the groups. A different study evaluated repeat LLR for HCC more than 1 cm from major vessels and found that repeat LLR and repeat OLR had comparable operating times and long-term outcomes but led to significantly less blood loss and less postoperative morbidity [51]. Along similar lines, several studies have shown that LLR facilitates future liver surgery. Laurent et al. reported that LLR facilitates salvage liver transplantation for HCC [52]. During liver transplantation for recurrent HCC (*n* = 19) or as a bridge to a transplant (*n* = 5), there was shorter hepatectomy time, shorter OR time, less EBL, and less pRBC transfused in patients who underwent prior LLR vs. OLR for HCC. 

LLR is increasingly widely accepted as the optimal surgical approach for patients with HCC and cirrhosis. There is strong international body of the literature, albeit mostly non-randomized, indicating improved short-term outcomes and similar long-term outcomes of LLR compared to OLR. Importantly, oncologic outcomes, both in terms of R0 resection rate, resection margin, and postoperative DFS, are not negatively impacted by the laparoscopic approach. Table 1 lists 15 comparative studies from 2001 to 2016 that provide 5-year overall survival (OS) data of LLR vs. OLR for HCC. There was no significant difference in 5-year OS in any of the studies. Moreover, LLR may be preferred even in selected patients with advanced cirrhosis or those requiring repeat LLR.

A number of meta-analyses on the topic have synthesized the available data and found that LLR offers consistent short-term advantages and leads to at least equivalent long-term oncologic outcomes. In fact, a recent, large meta-analysis by Kamarajah et al. [53] suggests that advances in minimally invasive techniques of liver resection may even confer an improved disease-specific survival, although these findings remain to be confirmed by subsequent studies. 

### 3.2. Colorectal Cancer Liver Metastases (CRLM)

Colorectal cancer is the third most common cancer worldwide [2]. In the last several decades, significant advances in imaging, surgical techniques, and chemotherapeutic regimens have improved survival. However, liver resection for patients with colorectal liver metastasis (CRLM) remains the only potentially curative option. CRLM is the second most common malignant indication for liver resection worldwide [1,54]. LLR for CRLM offers the benefits of reduced postoperative complications and shorter length of stay without a negative impact on long-term oncologic outcomes. Table 2 lists 16 comparative studies from 2009 to 2020 that provide 5-year OS data regarding LLR vs. OLR for CRLM. There was no significant difference in the 5-year OS in any of the studies.

A landmark publication that summarized the early experiences of 109 patients undergoing LLR for CRLM reported no postoperative mortality and a complication rate of 12% [55]. The median tumor size resected was 3.0 cm, and a good oncologic result was achieved, with 94.4% of patients having a negative (R0) margin and median margin of 10 mm. The 5-year OS and DFS were 50% and 43%, which is favorably comparable to OLR.

An Interesting study from France compared the outcomes of two specialized liver centers performing LLR and OLR [42]. Only first hepatectomies performed for CRLM were included, and the patients were comparable with respect to their preoperative prognostic characteristics. The types of resections did not differ between the groups. LLR and OLR had equivalent postoperative lengths of stay, transfusion rates, morbidity, and mortality. The LLR group included smaller tumors (30 mm vs. 41 mm) and had a higher R0 resection rate (87% vs. 72%) compared to LLR. There was no significant difference in 5-year OS (64% vs. 56%) or Recurrence-Free Survival (30% vs. 20%) between the LLR and OLR groups.

LLR and OLR had equivalent oncologic outcomes while estimated blood loss was lower and postoperative length of stay was shorter following LLR in a large Japanese study of patients with CRLM [37]. There was no difference in R0 resection rates or resection margin between the groups. It is important to note, however, that the majority of patients in either group had only one tumor and almost no patients had more than four. These tumors were mostly unilobar and smaller than 5 cm. The 5-year OS in this study was 70% and 68% for LLR and OLR, respectively. Similarly, a meta-analysis conducted by Schiffman and colleagues found that LLR confers a perioperative benefit in patients with CRLM limited to one or two tumors [56]. LLR led to lower blood loss and complication rates as well as shorter lengths of stay without a negative impact on 5-year DFS or OS. Finally, a European study seeking to understand the risk factors for R1 resection of CRLM identified nonanatomic resections, combined anatomic and nonanatomic resections, the number of lesions, and the size of tumors to be associated with higher incidence of microscopically positive margins. Interestingly, blood loss was only a risk factor for R1 resection for LLR and not OLR, while the Pringle maneuver was protective and the surgical approach itself was not a risk factor [57].

In the last several years, two randomized controlled trials compared LLR and OLR for CRLM and demonstrated that LLR leads to improvement in short-term morbidity without compromising long-term survival. In the OSLO-COMET trial, a total of 280 patients requiring parenchyma-sparing liver resection for CRLM were randomized to either an LLR or OLR group [29]. The study found that LLR was superior to OLR in terms of 30-day complications (19% vs. 31%, *p* = 0.021) and length of stay in a hospital (53 vs. 96 h, *p* < 0.001), while there was no difference in cost, blood loss, operation time, or resection margin. Importantly, a follow-up study from this clinical trial reported that the 5-year OS rates were equivalent at 54% and 55% for the LLR and OLR groups, respectively [58]. A subsequent LapOpHuva trial included 193 patients with resectable CRLM randomized scheduled to undergo LLR or OLR [28]. The patients in both groups had a similar number of tumors (a median of 1 in both the LLR and OLR groups, *p* = 0.89) and similar tumor sizes (median 4 vs. 3, *p* = 0.13). The primary outcome of interest was 90-day postoperative morbidity. In this study, LLR patients suffered fewer overall complications (11.5% vs. 23.7%, *p* = 0.025), had shorter lengths of stay (4 vs. 6 days, *p* < 0.001), and had similar long-term outcomes with a 5-year OS of 49.3% and 47.4%. There was no difference in operation time, blood loss, transfusion requirement, or R0 resection rate.

An Important additional advantage of LLR for CRLM is the ability to start systemic chemotherapy sooner than following OLR. Patients are able to receive adjuvant chemotherapy 2–3 weeks earlier after LLR than after OLR [59,60]. Postoperative complications significantly delayed chemotherapy after OLR in the corresponding OLR group but not after LLR in the LLR group. These results were corroborated by Mbah et al., who demonstrated shorter a length of stay and earlier initiation of systemic chemotherapy following laparoscopic major hepatectomies [61]. Although this study also included cholangiocarcinoma, carcinoid tumors, and others, over 60% of patients had CRLM. 

A number of meta-analyses have been performed in recent years investigating the effects of minimally invasive LLR and RLR on patients with CRLM (Table 4). Ozair et al.’s study published in 2022 compiled data from both randomized and non-randomized studies to determine whether minimally invasive techniques led to better outcomes in patients with CRLM both during staged and simultaneous resections [62]. The authors recapitulated the advantages of minimally invasive hepatic resection, including shorter length of stay, lower blood loss, and fewer complications, without compromising long-term oncologic outcomes. 

### 3.3. Robot-Assisted Liver Resection for CRLM

Robot-assisted liver resection (RLR) has been gaining in popularity as it has enabled the articulation of instruments, the elimination of tremors, and the improved three-dimensional visualization for fine dissections. However, limited data exist regarding the outcomes of using this approach for CRLM. A multicenter study compared RLR and LLR and found that the outcomes are generally similar between the groups [63]. The median tumor sizes were 2.5 and 2.4 cm for RLR and LLR, respectively; the majority of patients had one or two tumors and similar types of resection. There were no differences in postoperative complications, length of stay, intensive care unit admission, or resection margin, and the two groups had an equivalent 5-year OS at 61% and 60%. A different study compared RLR, LLR, and OLR for CRLM [64] and found that RLR was comparable to LLR in terms of postoperative complication rate, R0 resection, and operation time. The results of this study suggest that RLR may be associated with less blood loss than LLR or OLR, but the authors acknowledge that the number of RLR cases included is small and should be interpreted with caution.

### 3.4. Which Patients Should Be Selected for LLR vs. OLR?

The laparoscopic approach is preferred for many patients with HCC and CRLM but is not suitable for everyone. There are important tumor- and patient-related considerations to account for when evaluating a patient for LLR. Surgeons must know their comfort zones and limitations with respect to selecting suitable candidates for LLR. For hepatobiliary surgeons familiarizing themselves with LLR, solitary tumors in the left lateral section or anterolateral segments are the most straightforward. In cirrhotic settings, small tumors (<3 cm) near the capsule are preferred. As greater experience is gained, more challenging cases, including major hepatectomy, posterosuperior difficult segments, and select patients with Child’s B cirrhosis, can be undertaken. The learning curve with respect to LLR varies between surgeons and can depend on their prior expertise as well as the experience of the center in which they are practicing, but the data suggest that approximately 25–40 cases are required to undertake laparoscopic minor hepatectomy and 50–70 for laparoscopic major hepatectomy [65,66].

Difficulty scores have been proposed and validated to assist surgeons in terms of evaluating how challenging a given LLR case will be. The four most common difficulty scores are the Iwate, IMM (Institut Mutualiste Montsouris), Southampton, and Hasegawa scores, which are summarized in Table 3. A recent systematic review and meta-analysis of the available difficulty-scoring systems showed that all four scores were effective in terms of predicting the difficulty of LLR, and there was not a clearly superior scoring system [67].

The majority of studies examining the advantages of LLR for both HCC and CRLM have involved predominantly solitary tumors less than 5 cm. While it is technically feasible to perform LLR for larger lesions, the data in this regard are lacking, and an open approach may be more appropriate. Similarly, an open approach may be favored among patients with multifocal malignancy disease, as the benefits of LLR are less established.

In a meta-analysis of 610 patients undergoing LLR vs. OLR for CRLM, Schiffman et al. reported that well-matched case-cohort studies justified the use of LLR among patients with limited tumor burdens (1 or 2 metastases) [56]. The 5-year OS rate was 51% in the LLR group and 46% in the OLR group. Some studies have reported the results of LLR carried out on patients with up to four tumors; however, there is little high-quality evidence regarding the execution of LLR among patients with five or more tumors. In addition, laparoscopy may not be appropriate for patients with significantly elevated bleeding risk, adhesions due to previous operations, aberrant anatomy, decreased pulmonary compliance, elevated cardiovascular risk, or intracranial disease.

Regarding the LLR of HCC among cirrhotic patients, the majority of the studies reported concern Child’s A cirrhosis. However, several recent studies have indicated the safety of LLR in selected patients with Childs B cirrhosis [45,46].

**Table 3 cancers-15-01179-t003:** Summary of LLR Difficulty scores.

Score	Features	Scale
Iwate (Ban) [68] (Stratifies for degree of difficulty)	Total of 6 factors; Score 1–12 Location, size Near major vessel, Extent resection, HALS/Hybrid, Cirrhosis	Low (0–3) Intermediate (4–6) Advanced (7–9) Expert (10–12)
IMM (Kawaguchi) [48] (Stratifies for risk of morbidity)	OR time (< or ≥190 min) EBL (< or ≥100 mL) Conversions (< or ≥4.2%)	I—Low (wedge/LLS) II—Intermediate (anterolateral segm, Left hepatectomy) III—High (post/superior segm, R posterior sectionectomy, R, central or extended L/R hepatectomy)
Hasegawa [69] (Stratified for surgical difficulty)	Extent of resection Tumor location Obesity (BMI > 30) Platelet count (<100)	Low (0–1) 136 min Medium (2–3) 225 min High (≥4) 324 min
Southampton (Halls) [70] (Stratified for intraoperative complications)	5 factors; Score 1–15 Neoadjuvant chemo 0/1 Prior open liver resection 0/5 Lesion type (B/M) 0/2 Lesion size (<3, 3–5, >5) 0/2/3 Extent of resection 0/2/4	Low (0–2) Moderate (3–5) High (6–9) Extremely High (10–15)

**Table 4 cancers-15-01179-t004:** Summary of meta-analyses published on the topic of laparoscopic liver resection (LLR), robotic liver resection (RLR), and open liver resection (OLR). The studies are listed in order of year of publication.

Study	Year	Journal	# of Patients	Patient Population
Kelly et al. [71]	2022	Ir J Med Sci	3095	CRLM
Wang et al. [72]	2022	Int J Surg	1346	HCC in elderly Patients
Aboudou et al. [73]	2022	J Clin Med	1783	RLR vs. LLR
Wang et al. [74]	2022	Front Oncol	1861	Elderly Patients
Murtha-Lemekhova et al. [75]	2022	Cancers (Basel)	529	RLR vs. LLR
Kamarajah et al. [53]	2022	Scand J Surg	13,731	Minimally invasive hepatectomy (RLR + LLR) for HCC
Ozair et al. [62]	2022	Surg Endosc	2800	CRLM
Ciria et al. [76]	2022	J Hepatobiliary Pancreat Sci	2728	RLR vs. LLR
Rahimli et al. [77]	2022	Cancers (Basel)	1530	RLR vs. LLR
Hao et al. [78]	2022	Front Oncol	945	Repeat hepatectomy for recurrent HCC
Hajibandeh et al. [79]	2022	Langenbecks Arch Surg	319	RLR vs. LLR for left lateral sectionectomy
Wang et al. [80]	2022	Front Surg	541	LLR vs. OLR for right posterior sectionectomy
Kabir et al. [44]	2021	Br J Surg	1618	HCC in patients with cirrhosis
Haney et al. [81]	2021	HPB (Oxford)	1457	Benign + malignant
Kamarajah et al. [82]	2021	Scand J Surg	2630	RLR vs. LLR
Hu et al. [83]	2021	Asian J Surg	1093	RLR vs. LLR
Ding et al. [84]	2021	Langenbecks Arch Surg	237	Caudate Lobe Tumors
Ziogas et al. [85]	2021	Surg Endosc	525	RLR vs. LLR
Mohamedahmed et al. [86]	2021	Updates Surg	1762	Malignant Tumors in Elderly Patients
Chen et al. [87]	2021	Front Oncol	2728	Recurrent Liver Tumors
Wang et al. [88]	2021	Medicine (Baltimore)	751	RLR vs. LLR for minor hepatectomy
Coletta et al. [89]	2021	Int J Med Robot	485	RLR vs. LLR for major hepatectomy
Sun et al. [90]	2021	Hepatol Int	8905	HCC
Zhao et al. [91]	2021	Updates Surg	2999	RLR vs. LLR vs. OLR
Pan et al. [92]	2021	Front Oncol	1975	HCC in patients with cirrhosis
Pan et al. [93]	2020	World J Surg Onc	616	Simultaneous resection of colon cancer and CRLM
De’Angelis et al. [94]	2020	PloS One	1160	CRLM in elderly patients
Syn et al. [95]	2020	Ann Surg	3148	CRLM
Zhang et al. [96]	2020	PloS One	3544	RLR vs. LLR
Hajibandeh et al. [97]	2020	Surg Laparosc Endosc Percutan Tech	1023	Posterosuperior Segment Tumors
Liang et al. [98]	2020	Int J Surg	767	Repeat liver resection for malignancy
Rubinkiewicz et al. [99]	2020	Wideochir Inne Tech Maloinwazyjne	1196	Posterolateral Segment Tumors
Hong et al. [100]	2020	Medicine (Baltimore)	467	Right Hepatectomy Only
Coletta et al. [101]	2020	J Laparoendosc Adv Surg Tech A	1321	Patients with cirrhosis and portal hypertension
Ciria et al. [102]	2020	Surg Endosc	3308	CRLM
Solaini et al. [103]	2020	J Laparoendosc Adv Surg Tech A	833	LLR vs. OLR for anatomic resection of HCC
Xing et al. [104]	2020	Eur J Gastroenterol Hepatol	1459	HCC in patients with cirrhosis
Guan et al. [105]	2019	Asian J Surg	938	RLR vs. LLR
Notarnicola et al. [106]	2019	Surg Endosc	1025	Elderly Patients
Peng et al. [107]	2019	Medicine (Baltimore)	1232	Repeat Liver Resections
Wang et al. [108]	2019	BMC Cancer	1173	Major liver resections for HCC
Zheng et al. [109]	2019	Surg Endosc	788	Posterosuperior Tumors
Witowski et al. [110]	2019	Surg Endosc	5100	HCC
Yin et al. [111]	2019	Medicine (Baltimore)	1163	Left Hepatectomy Only
Macacari et al. [112]	2019	Int J Surgery	3415	Left Lateral Sectionectomy Only
Xiangfei et al. [113]	2019	Surg Endosc	6812	HCC
Ye et al. [114]	2019	World J Gastroenterol	502	Minimally Invasive vs. OLR for simultaneous colorectal cancer and CRLM resection
Ciria et al. [115]	2019	Ann Surg Onc	8454	HCC
Peng et al. [116]	2019	Surg Oncol	443	LLR vs. OLR for recurrent liver tumors
Cai et al. [117]	2019	Surg Endosc	335	LLR vs. OLR for recurrent HCC
Shang et al. [118]	2019	J Laparoendosc Adv Surg Tech A	3897	HCC
Liu et al. [119]	2018	J Lap Adv Surg Tech A	638	Segment I, Iva, VII, and VIII lesions
Wang et al. [120]	2018	Clin Res Hepatol Gastroenterol	1573	Cirrhosis
Jiang et al. [121]	2018	Hepatol Res	5889	HCC
Jin et al. [122]	2018	Surg Oncol	554	Benign + malignant
Hu et al. [123]	2018	Asian J Surg	1389	RLR vs. LLR
Machairas et al. [124]	2018	Surg Laparosc Endosc Percutan Tech	531	Posterior Superior Segment Tumors
Chen et al. [125]	2018	Surg Laparosc Endosc Percutan Tech	780	Major Hepatectomy for HCC
Zacharoulis et al. [126]	2018	HPB (Oxford)	2640	Left Lateral Sectionectomy Only
Chen et al. [127]	2018	Can J Gastroenterol Hepatol	830	HCC
Kasai et al. [128]	2018	Surgery	917	Major Hepatectomy Only
Yin et al. [129]	2018	Int J Surg	647	Posterosuperior Segment Tumors
Sotiropoulos et al. [43]	2017	Updates Surg	5203	HCC
Komorowski et al. [130]	2017	Arch Med Sci	1976	Benign + malignant
Zhang et al. [131]	2017	Int J Surg	2259	CRLM
Xie et al. [132]	2017	Sci Rep	4697	CRLM
Sotiropoulos et al. [133]	2017	J BUON	851	HCC
Xu et al. [134]	2017	Scand J Gastroenterol	1130	Major Hepatectomy
Cheng et al. [135]	2017	J Surg Res	4591	CRLM
Tian et al. [136]	2016	Oncotarget	1679	CRLM
Nota et al. [137]	2016	HPB (Oxford)	363	RLR only
Qiu et al. [138]	2016	Surg Endosc	537	RLR vs. LLR
Hallet et al. [139]	2016	Hepatobiliary Surg Nutr	2017	CRLM
Schiffman et al. [56]	2015	Surgery	610	CRLM
Morise et al. [140]	2015	J Hepatobiliary Pancreat Sci	2466	HCC and chronic liver disease
Jackson et al. [141]	2015	JSLS	3702	RLR vs. LLR vs. OLR
Montalti et al. [142]	2015	World J Gastroenterol	694	RLR vs. LLR
Luo et al. [143]	2014	J Lap Adv Surg Tech	624	CRLM
Wei et al. [144]	2014	PloS One	975	CRLM
Parks et al. [145]	2014	HPB	1002	HCC + mCRC
Twaij et al. [146]	2014	W J Gastro	420	HCC in patients with cirrhosis
Rao et al. [147]	2013	Cochrane review	no RCT	N/A
Yin et al. [148]	2013	Ann Surg Onc	1238	HCC
Zhou et al. [149]	2013	BMC Surg	695	CRLM
Rao et al. [150]	2012	Am J Surg	2466	Benign + malignant
Xiong et al. [151]	2012	W J Gastro	550	HCC
Li et al. [152]	2012	Hepatol Res	627	HCC
Rao et al. [153]	2012	Surgeon	700	Malignant
Mirnezami et al. [154]	2011	HPB	1678	Benign + malignant
Rao et al. [155]	2011	Surg Endosc	245	Benign + malignant
Zhou et al. [156]	2011	DDS	494	HCC
Fancellu et al. [157]	2011	J Surg Res	590	HCC
Mizuguchi et al. [158]	2011	Surg Today	485	HCC
Croome et al. [159]	2010	Arch Surg	1890	Benign + malignant
Simillis et al. [160]	2007	Surgery	403	Benign + malignant

## 4. Conclusions

LLR has been performed on thousands of patients with HCC or CRLM worldwide. The majority of the studies show that LLR offers patients reduced perioperative morbidity, shorter length of stay, and faster recovery with equivalent long-term oncologic outcomes. Among patients with cirrhosis, LLR results in decreased liver decompensation compared to OLR and growing evidence suggests that it can be safely performed in patients with Childs B disease. Two randomized controlled trials have demonstrated that LLR for CRLM leads to shorter length of stay and less morbidity while being comparable to OLR in terms of R0 resection rate, disease-free survival, and overall survival. In conclusion, utilizing LLR offers significant advantages to patients with HCC and CRLM, without posing concerns regarding whether oncologic outcomes will be compromised.

## Figures and Tables

**Table 1 cancers-15-01179-t001:** Summary of studies reporting 5-year Overall Survival (OS) following Laparoscopic Liver Resection vs. Open Liver Resection in patients with Hepatocellular Carcinoma (HCC). The studies are listed in order of year of publication. RCT—randomized controlled trial; PSM—propensity-score-matched.

Study	Study Design/# of Patients (LLR/OLR)	Year	Journal	Country	Overall Survival % (LLR/OLR)	*p*
El-Gendi et al. [10]	RCT 25/25	2018	J Laparoendosc Adv Surg Tech A	Egypt	59/54 (3-year DFS)	NS
Cheung et al. [11]	PSM, Retrospective 110/330	2016	Ann Surg	Hong Kong	84/67	NS
Chang et al. [12]	Retrospective 30/30	2016	Ann Acad Med Singapore	Singapore	59/65	NS
Takahara et al. [13]	PSM, Retrospective 387/387	2015	J Hepatobiliary Pancreat Sci	Japan	77/71	NS
Kim et al. [14]	PSM, Retrospective 29/29	2014	Surg Endosc	Korea	92/88	NS
Cheung et al. [15]	Retrospective, 32/64	2013	Ann Surg	Hong Kong	77/57	NS
Kim et al. [16]	Retrospective, 26/29	2011	J Korean Surg Soc	Korea	57/56	NS
Hu et al. [17]	Retrospective, 30/30	2011	World J Gastroenterol	China	50/53	NS
Lee et al. [18]	Retrospective, 33/50	2011	World J Surg	Hong Kong	76/76	NS
Truant et al. [19]	Retrospective, 36/53	2011	Surg Endosc	France	70/46	NS
Ker et al. [20]	Retrospective, 116/208	2011	Int J Hepatol	Taiwan	62/72	NS
Tranchart et al. [21]	Retrospective, 42/42	2010	Surg Endosc	France	60/47	NS
Endo et al. [22]	Retrospective, 10/11	2009	Surg Laparosc Endosc Percutan Tech	Japan	57/48	NS
Sarpel et al. [23]	Retrospective, 20/56	2009	Ann Surg Oncol	USA	95/75	NS
Cai et al. [24]	Retrospective, 31/31	2008	Surg Endosc	China	50/51	NS
Kaneko et al. [25]	Retrospective, 30/28	2005	Am J Surg	Japan	61/62	NS
Shimada et al. [26]	Retrospective, 17/38	2001	Surg Endosc	Japan	50/38	NS

**Table 2 cancers-15-01179-t002:** Summary of studies reporting 5-year Overall Survival (OS) following Laparoscopic Liver Resection vs. Open Liver Resection among patients with Colorectal Liver Metastases (CRLM). The studies are listed in order of year of publication. RCT—randomized controlled trial; PSM—propensity-score-matched.

Study	Study Design/# of Patients (LLR/OLR)	Year	Journal	Country	Overall Survival % (LLR/OLR)	*p*
Efanov et al. [27]	PSM, Retrospective, 20/20	2021	Surg Endosc	Russia	78/63	NS
Robles-Campos et al. [28]	RCT, 96/97	2019	Surg Endosc	Spain	49/47	NS
Fretland et al. [29]	RCT, 133/147	2018	Ann Surg	Norway	54/55	NS
Gourmand et al. [30]	PSM, Retrospective, 43/121	2018	HPB (Oxford)	USA	81/68	NS
Lewin et al. [31]	PSM, Retrospective, 140/122	2016	HPB (Oxford)	Australia	54/63	NS
Cipriani et al. [32]	PSM, Retrospective, 133/133	2016	Br J Surg	UK	63/64	NS
Lin et al. [33]	PSM, Retrospective, 36/36	2015	Inr J Colorectal Dis	China	51/55	NS
Hasegawa et al. [34]	Retrospective, 100/68	2015	Surgery	Japan	57/49	NS
de’Angelis et al. [35]	PSM, Retrospective, 52/52	2015	J Laparoendosc Adv Surg Tech A	France	76/62	NS
Allard et al. [36]	Retrospective, 73/73	2015	Ann Surg	France	78/75	NS
Beppu et al. [37]	PSM, Retrospective 171/342	2015	J Hepatobiliary Pancreat Sci	Japan	70/68	NS
Montalti et al. [38]	Retrospective, 57/57	2014	Eur J Surg Oncol	Belgium	60/65	NS
Iwahashi et al. [39]	Retrospective, 21/21	2014	Surg Endosc	France	42/51	NS
Cannon et al. [40]	Retrospective, 35/140	2012	Surgery	USA	36/42	NS
Topal et al. [41]	Retrospective, 20/20	2012	Surg Endosc	Belgium	48/46	NS
Castaing et al. [42]	Retrospective, 60/60	2009	Ann Surg	France	64/56	NS

## Data Availability

No new data were created or analyzed in this study. Data sharing is not applicable to this article.

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
