# Peer review of "Outcomes and Patient Selection in Laparoscopic vs. Open Liver Resection for HCC and Colorectal Cancer Liver Metastasis"

_cancers, 2023, doi:10.3390/cancers15041179_

Round 1
Reviewer 1 Report
One recommendation only:
Usually, in the selection of laparoscopic and open hepatectomy, there is bias in tumor burden and location. I think it is necessary to analyze and describe in your cited paper.
Author Response
Response: Thank you for this insightful comment, and we have added to the discussion:
“Majority of studies examining the advantages of LLR for both HCC and CRLM included predominantly solitary tumors less than 5 cm. While it is technically feasible to perform LLR for larger lesions, data on this question is lacking and an open approach may be more appropriate. Similarly, open approach may be favored in patients with multifocal malignancy disease as benefits of LLR are less established.”
Reviewer 2 Report
Dear Editor,
Alvikas et al. submitted the manuscript “Patient Selection and Outcomes of Laparoscopic vs. Open Liver Resection for HCC and Colorectal Cancer Liver Metastasis (CRLM)”. The authors performed a review of the current literature on the outcomes of laparoscopic liver resection (LLR) versus open liver resection (OLR) for patients with HCC and CRLM. Furthermore, the authors included a short paragraph on Robot-assisted liver resection (RLR) for CRLM and described an approach to select patients best suited for LLR.
The new evidence gained by this study is very limited due to previous systematic reviews and meta-analysis
(Sotiropoulos G.C. et al. Meta-analysis of laparoscopic vs open liver resection for hepatocellular carcinoma. Updates Surg 69, 291–311 (2017). https://doi.org/10.1007/s13304-017-0421-4;
Kelly ME et al. Open versus laparoscopic liver resection of colorectal metastases: a meta-analysis of matched patient populations. Ir J Med Sci. 2022 Aug;191(4):1531-1538. doi: 10.1007/s11845-021-02780-3. Epub 2021 Sep 17. Erratum in: Ir J Med Sci. 2022 Aug 5;: PMID: 34535883.,
Xiong JJ et al. Meta-analysis of laparoscopic vs open liver resection for hepatocellular carcinoma. World J Gastroenterol 2012; 18(45): 6657-6668 [PMID: 23236242 DOI: 10.3748/wjg.v18.i45.6657],
Twaij A et al. Laparoscopic vs open approach to resection of hepatocellular carcinoma in patients with known cirrhosis: systematic review and meta-analysis. World J Gastroenterol. 2014 Jul 7;20(25):8274-81. doi: 10.3748/wjg.v20.i25.8274. PMID: 25009403; PMCID: PMC4081703.,
Rao A et al. Laparoscopic vs. open liver resection for malignant liver disease. A systematic review. Surgeon. 2012 Aug;10(4):194-201. doi: 10.1016/j.surge.2011.06.007. Epub 2011 Jul 30. PMID: 22818276.,
Pan L et al. Laparoscopic procedure is associated with lower morbidity for simultaneous resection of colorectal cancer and liver metastases: an updated meta-analysis. World J Surg Oncol. 2020 Sep 21;18(1):251. doi: 10.1186/s12957-020-02018-z. PMID: 32958079; PMCID: PMC7507629.,
de'Angelis N et al. Surgical and regional treatments for colorectal cancer metastases in older patients: A systematic review and meta-analysis. PLoS One. 2020 Apr 22;15(4):e0230914. doi: 10.1371/journal.pone.0230914. Erratum in: PLoS One. 2021 Apr 27;16(4):e0251005. PMID: 32320417; PMCID: PMC7176093.).
As this is only a review and not a systematic review or meta-analysis, the overall results can only be interpreted to a limited extent and other studies (as mentioned above) provide higher evidence, limiting the impact of this paper. Furthermore, the quality of the manuscript would be improved if focusing only on one or two questions and not three.
Major Comments:
1. The results were not surprising because previous studies and meta-analyses already showed the similar results. What are the new findings and different points from previous studies?
2. Please consider a revision of the manuscript according to the PRISMA Checklist/Guideline and PRISMA for Abstracts Checklists. Even though this is not a systematic review, vital structural and methodical key information are missing (e.g.: method section/conclusion)
Minor Comments:
3. When giving statements or citing numbers, please provide references and sources for statements.
4. Consider shortening the reference list.

Author Response
Response: Thank you for this insight and the useful references. While we agree that meta-analysis can provide a high level of evidence, the objective of the present manuscript was a narrative review, as opposed to meta-analysis, summarizing current data on LLR and OLR for HCC and CRLM. In response to the comment, however, we include an additional and novel Table 4 which highlight and summarizes the vast number (over 90) meta-analyses that have been published on this topic. We also include this text:
“A large number of meta-analyses have been performed in recent years investigating the effects of minimally invasive LLR and RLR on patients with CRLM and HCC (Table 4). Ozair et al. study published in 2022 compiled data from both randomized and non-randomized studies to see whether minimally invasive techniques led to better outcomes in patients with CRLM, both during staged and simultaneous resections.34 Authors re-demonstrated the advantages of minimally invasive hepatic resection, including shorter length of stay, lower blood loss and fewer complications, without compromising long-term oncologic outcomes. “
“A number of meta-analyses on the topic have synthesized the available data and found that LLR has consistent short-term advantages and lead to at least equivalent long-term oncologic outcomes. In fact, a recent large meta-analysis by Kamarajah et al.22 suggests that advances in minimally invasive techniques of liver resection may even confer an improved disease-specific survival, although these findings remain to be confirmed by subsequent studies. “
Reviewer 3 Report
Dear Authors,
In your review, you very accurately describe the current status of laparoscopic liver resection for HCC and CRLM. From a medical point of view, all statements are correct.
Nevertheless, I cannot recommend publication of the manuscript in Cancers for the following reasons:
1) Methods
- How were the studies selected for your review?
- Did the selection follow a systematic approach (PRISMA)?
- What search terms did you use?
- What time periods were screened?
- What databases were searched?
- Were there inclusion and exclusion criteria?
- here a flowchart for study selection would be useful
2) HCC and CRLM are two completely different diseases, as you also correctly described. It does not make sense, in my opinion, to cover both diseases in one review.
3) Tables
- the study designs (prospective/retrospective etc) should be mentioned
- inclusion and exclusion criteria of the study should be mentioned (minor resections? major resections?)
- why are there studies from 2001 - 2018 in table 1 and from 2009 - 2021 in table 2 (see also point 1: methods)?
4) What (new) insights does a reader get from your review that are not already known? In my opinion, it is common knowledge of a hepatobiliary surgeon.
5) Further points
- In your text, you present successive study results on the topic. Results are strung together. A common thread is missing. I would recommend to comment on different parameters one after the other, e.g.: HCC: a) blood loss: study A by XY reported reduced blood loss by LLR....and so on; b) bile leaks; c) operating time; d) surgical complications according to Clavin-Dindo; e) resection margins; f) oncological survival (DFS, OS) etc.
- In line 155, you list 14 studies for Table 2, but then Table 2 lists 16 studies.
- Lines 229-232: what should be the annual number of cases for a laparoscopic liver surgeon or institution to perform laparoscopic liver surgery with high patient safety?
- Lines 25-26 and line 5: you mention robotic liver surgery, which you would like to address in your review. Unfortunately, other than in the abstract and introduction, robotic is not addressed in the entire text.
Author Response
1) Methods
- How were the studies selected for your review?
- Did the selection follow a systematic approach (PRISMA)?
- What search terms did you use?
- What time periods were screened?
- What databases were searched?
- Were there inclusion and exclusion criteria?
- here a flowchart for study selection would be useful
Response: We apologize for this oversight and have added a Methods section addressing the questions above. We include this in our text:
“We performed a literature review of studies published in Pubmed database from years 2001 – 2022, and used search terms “laparoscopic liver resection, minimally invasive liver resection, robotic liver resection, HCC, colorectal cancer liver metastasis, and meta-analysis. We excluded studies on benign disease, as well as studies that focused on malignancies other than HCC and CRLM. From these studies identified, we performed a literature review focusing on outcomes of laparoscopic liver resections for CRLM and HCC. “
2) HCC and CRLM are two completely different diseases, as you also correctly described. It does not make sense, in my opinion, to cover both diseases in one review.
Response: We included both HCC and CRLM because these are the two most common indications for LLR. Two large world reviews showed that about 50% of LLR are for HCC and 25% for CRLM. So the purpose of this review was to provide a clearinghouse of studies including new table 4 which contains 95 meta-analyses studies that provide the reader with the most up-to-date review of LLR for CRLM and HCC.
3) Tables
- the study designs (prospective/retrospective etc) should be mentioned
- inclusion and exclusion criteria of the study should be mentioned (minor resections? major resections?)
- why are there studies from 2001 - 2018 in table 1 and from 2009 - 2021 in table 2 (see also point 1: methods)?
Response: Table includes three studies from 2001, 2005, and 2008, and the remainder are 2009 and beyond. Table 2 includes 2009 to 2021 studies because we focused on studies that gave 5-yr OS of LLR vs OLR for CRLM. There were no studies published prior to 2009 that had 5-yr OS data.
4) What (new) insights does a reader get from your review that are not already known? In my opinion, it is common knowledge of a hepatobiliary surgeon.
Response: The field is expanding rapidly, and it can be hard to keep up with literature. The new Table 4 provides a summary of over 90 meta-analyses published on the topic of laparoscopic liver resection (LLR), robotic liver resection (RLR) and open liver resection (OLR). This should provide new and useful summary for the hepatobiliary surgeon.
5) Further points
- In your text, you present successive study results on the topic. Results are strung together. A common thread is missing. I would recommend to comment on different parameters one after the other, e.g.: HCC: a) blood loss: study A by XY reported reduced blood loss by LLR....and so on; b) bile leaks; c) operating time; d) surgical complications according to Clavin-Dindo; e) resection margins; f) oncological survival (DFS, OS) etc.
- In line 155, you list 14 studies for Table 2, but then Table 2 lists 16 studies.
Response: We have corrected the manuscript.
- Lines 229-232: what should be the annual number of cases for a laparoscopic liver surgeon or institution to perform laparoscopic liver surgery with high patient safety?
Response: Several studies show that the learning curve for LLR for minor hepatectomy is 25-40 cases, and for major hepatectomy 50-70 cases. WE have added this to the text with 2 new citations.
- Lines 25-26 and line 5: you mention robotic liver surgery, which you would like to address in your review. Unfortunately, other than in the abstract and introduction, robotic is not addressed in the entire text.
Response: Thank you for the comment. We have included a paragraph on robotic liver resection (RLR) for CRLM, and have added a new Table 4 summarizing 75 meta-analyses and systematic reviews that include many studies comparing RLR vs LLR and RLR vs OLR.
Reviewer 4 Report
Thank you for the opportunity to review this review article on laparoscopic liver resection. The authors summarise the current literature in a concise and structured way.
I only have two comments, see below.
On row 198-200 the authors write: "In this study, LLR patients suffered fewer overall complications (23.7% vs 11.5%, p=0.025), had shorter length of stay (4 vs 6 days, p<0.001) and had similar long-term outcomes with 5-year OS of 49.3% and 47.4%." A few sentences earlier LLR results are presented first, here they are reported last, then first and then it is unclear. Please check the manuscript to make certain that the order is consistent as this makes it easier to read.
On line 237 difficulty scores are listed, in the references publication there are 5 scores, not four as the authors list. Why was the Ban-score omitted? It does not seem to be the same as Iwate as is suggested in table 3.
Author Response
I only have two comments, see below.
On row 198-200 the authors write: "In this study, LLR patients suffered fewer overall complications (23.7% vs 11.5%, p=0.025), had shorter length of stay (4 vs 6 days, p<0.001) and had similar long-term outcomes with 5-year OS of 49.3% and 47.4%." A few sentences earlier LLR results are presented first, here they are reported last, then first and then it is unclear. Please check the manuscript to make certain that the order is consistent as this makes it easier to read.
Response: Thank you, we’ve revised the manuscript as suggested.
On line 237 difficulty scores are listed, in the references publication there are 5 scores, not four as the authors list. Why was the Ban-score omitted? It does not seem to be the same as Iwate as is suggested in table 3.
Response: Iwate score is the revised (and renamed) version of the original Ban-score, therefore we felt they could be grouped together.
Round 2
Reviewer 2 Report
accept
Author Response
Response: Thank you. Minor spell check performed, as suggested.
Reviewer 3 Report
Overall impression
This is a literature review on the value of laparoscopic liver resection for patients with CRLM and HCC. It is not a systematic review or meta-analysis. The authors' statements are understandable. However, as I noted in my initial review, the manuscript nevertheless lacks systematicity and structure (e.g. methods, see below).
The focus of the results is the comparison of the "outcome", and the patient selection is only a small part of the evaluation. Therefore, already the title should be rephrased: "Outcomes and Patient Selection of Laparoscopic vs. Open Liver Resection for HCC and Colorectal Cancer Liver Metastasis".
Methods
The search strategy was roughly described with indication of the search terms. How many hits matches did you get, and how many studies were excluded? Here, as suggested in my first review, a flowchart for study selection would be recommended. Even though this is not a systematic review, the description of the methods section is too brief. You stated that studies without 5-year survival data were excluded. This is not mentioned in the methods section, for example.
Results
- First, the results of the literature search should be described, e.g., how many studies you finally included in your review (studies on CRLM, studies on HCC, etc.)
- "Majority of studies indicate that LLR..." This is a summary of your results and therefore fits more in "Conclusion".
- 3.1 HCC: the first sentences "Primary liver cancer is the third leading cause...in this patient population." belong in the introduction, because it generally describes HCC
- "...but significantly longer operation time (294 vs 271 min, pp=0.025)" pp please correct
- 3.2 Colorectal Cancer Liver Metastases (CRLM): "Colorectal cancer is...worldwide": this paragraph belongs in the introduction and is not a result
- Some statements in the results section are not supported with references: "However, several recent studies have expanded LLR to patients with Child's B cirrhosis." Which recent studies do you refer to?
- "study designs" should also be stated in the tables
Table 4:
- 95 meta-analyses are mentioned, but not cited in “references” (Kasai et al. 2018; Nota et al. 2016, etc.) - this should be made up, as in Tables 1, 2, and 3
- the listing of studies in the table is in order of year of publication - this should be mentioned
- what information does the reader get from table 4, other than that 95 meta-analyses were found on LLR, RLR, and OLR?
Author Response
Overall impression
This is a literature review on the value of laparoscopic liver resection for patients with CRLM and HCC. It is not a systematic review or meta-analysis. The authors' statements are understandable. However, as I noted in my initial review, the manuscript nevertheless lacks systematicity and structure (e.g. methods, see below).
The focus of the results is the comparison of the "outcome", and the patient selection is only a small part of the evaluation. Therefore, already the title should be rephrased: "Outcomes and Patient Selection of Laparoscopic vs. Open Liver Resection for HCC and Colorectal Cancer Liver Metastasis".
Response: Title was rephrased to “Outcomes and Patient Selection in Laparoscopic vs. Open Liver Resection for HCC and Colorectal Cancer Liver Metastasis”, as suggested.
Methods
The search strategy was roughly described with indication of the search terms. How many hits matches did you get, and how many studies were excluded? Here, as suggested in my first review, a flowchart for study selection would be recommended. Even though this is not a systematic review, the description of the methods section is too brief. You stated that studies without 5-year survival data were excluded. This is not mentioned in the methods section, for example.
Response: Thank you for the helpful suggestion. We include the number of hits our search had and how many studies we included (see next comment below). We also added to the methods section that we excluded studies without 5-year overall survival (OS) for tables 1 and 2. Given this is not a systematic review, we politely decline including a PRISMA flowchart as we feel that may mislead the reader to believe this publication is a systematic review.
Results
- First, the results of the literature search should be described, e.g., how many studies you finally included in your review (studies on CRLM, studies on HCC, etc.)
Response: We added the number of studies that we included to the results section:
We include 17 studies on HCC (Table 1), and 16 studies on CRLM (Table 2). Our search of meta-analysis on LLR identified 224 results of which we identified 95 meta-analyses that have focused on outcomes LLR and compared it to OLR or RLR in different clinical settings (Table 4).
- "Majority of studies indicate that LLR..." This is a summary of your results and therefore fits more in "Conclusion".
Response: We deleted this comment from the results section and moved it to the conclusion:
“The majority of the studies show that LLR offers patients reduced perioperative morbidity, shorter length of stay and faster recovery with equivalent long-term oncologic outcomes.”
- 3.1 HCC: the first sentences "Primary liver cancer is the third leading cause...in this patient population." belong in the introduction, because it generally describes HCC
Response: This text was moved to the Introduction.
- "...but significantly longer operation time (294 vs 271 min, pp=0.025)" pp à please correct
Response: Corrected.
- 3.2 Colorectal Cancer Liver Metastases (CRLM): "Colorectal cancer is...worldwide": this paragraph belongs in the introduction and is not a result
Response: This text was moved to the Introduction.
- Some statements in the results section are not supported with references: "However, several recent studies have expanded LLR to patients with Child's B cirrhosis." Which recent studies do you refer to?
Response: The 2 references below are summarized and cited:
- Beard RE, Wang Y, Khan S, Marsh JW, Tsung A, Geller DA. Laparoscopic liver resection for hepatocellular carcinoma in early and advanced cirrhosis. HPB. 2018;20(6):521-529. doi:10.1016/j.hpb.2017.11.011
- Troisi RI, Berardi G, Morise Z, et al. Laparoscopic and open liver resection for hepatocellular carcinoma with Child-Pugh B cirrhosis: multicentre propensity score-matched study. Br J Surg. 2021;108(2):196-204. doi:10.1093/bjs/znaa041
Response:
- "study designs" should also be stated in the tables
Response: Study design are now included.
Table 4:
- 95 meta-analyses are mentioned, but not cited in “references” (Kasai et al. 2018; Nota et al. 2016, etc.) - this should be made up, as in Tables 1, 2, and 3
Response: References included.
- the listing of studies in the table is in order of year of publication - this should be mentioned
Response: Included.
- what information does the reader get from table 4, other than that 95 meta-analyses were found on LLR, RLR, and OLR?
Response: Table 4 provides an overview on the most recent meta-analyses performed on the topic, including comparisons in specific clinical situations, such as resections of difficult lesions, laparoscopic surgery in elderly patients, laparoscopy in patients with cirrhosis as well as robotic surgery. It also provides the total number of patients in the studies.
Round 3
Reviewer 3 Report
Thank you for the renewed opportunity to review. The authors have revised the manuscript in detail on all points. In particular, the methodology of the literature search and inclusion/exclusion criteria are now more comprehensible. There are no more reservations from my side against a publication - accept!